



# Linear Ice Fraction: Sea Ice Concentration Estimates from the ICESat-2 Laser Altimeter

Christopher Horvat[1,2], Ellen Buckley[3], Madelyn Stewart[4], Poom Yoosiri[2], and Monica M. Wilhelmus[3]

[1]Department of Physics, The University of Auckland, Auckland, NZ
[2]Department of Earth, Environmental, and Planetary Sciences, Brown University, Providence, USA
[3]Center for Fluid Mechanics, School of Engineering, Brown University, Providence, RI, US
[4]Department of Earth and Planetary Science, Yale University, New Haven, USA

**Correspondence:** Christopher Horvat (horvat@brown.edu)

**Abstract.** Sea ice coverage is a key indicator of changes in the global climate. Estimates of sea ice area and extent are primarily derived from satellite measurements of surface microwave emissions, from which local sea ice concentration (SIC) is derived. Passive microwave (PM) satellite sensors remain the sole global product for understanding SIC variability. Using a dataset of more than 27,000 high-resolution airborne optical images, we first examine biases in commonly-used products that emerge
from challenges in sampling thin sea ice fractures and melt ponds on the sea ice surface. We show that the ICESat-2 (IS2) laser altimeter effectively samples these surface features and we develop a new, independent SIC product, which we term the Linear Ice Fraction (LIF). On monthly timescales, we show using an emulator that the LIF product offers an independent estimate of sea ice concentration over 60% of the Arctic sea ice cover with similar-or-better error qualities compared to PM data. IS2 and its high-precision measurements of the sea ice surface should be considered for augmenting PM-SIC measurements in the
future.

## 1 Introduction

Sea ice concentration (SIC), the fraction of an ocean area covered by sea ice, is critically important for understanding Earth's climate variability. Since the late 1970s, SIC is estimated globally using passive microwave (PM) satellites at both hemispheres. Numerous algorithms have been developed (at least 11, (Kern et al., 2019)), which convert surface radiative properties to
gridded SIC on time scales from days to months. PM-derived SIC is a standard for assessing sea ice state and change (Meredith et al., 2022). Increasingly, SIC products are assimilated into state-of-the-art forecast and climate models at both hemispheres (Mazloff et al., 2010; Sakov et al., 2012; Massonnet et al., 2015; Verdy and Mazloff, 2017; Zhang et al., 2018; Fritzner et al., 2019; Zhang et al., 2021). However, the these SIC products are limited by PM resolution, which does not resolve small scale features present in the sea ice cover.

Sea ice is a heterogeneous, fractured mosaic of solid floes or plates ranging in size from meters to hundreds of kilometers and whose surface is comprised of some combination of ice, snow, and meltwater. Cracks in the sea ice, known as leads, are narrow in width and vary over length scales of kilometers to hundreds of kilometers and open and close on timescales of minutes to weeks (Bouillon and Rampal, 2015; Hutter et al., 2019; Ólason et al., 2020; Hutter and Losch, 2020). PM-SIC uncertainty





can arise from the occurrence of leads that are not easily observed with PM satellites given their near-linear geometry and variability. When examining 11 different PM-SIC products in regions with near-100% SIC in winter, Kern et al. (2019) found systematic algorithmic differences between products that range from -1.1% to 3.5%. While these differences are small in terms of the overall SIC, air-sea exchange in leads is an important source of ocean mixing and energy in winter. A second, larger discrepancy in PM-SIC comes in summer, when PM-SIC estimates vary up to 35% (Kern et al., 2020). This is due to the presence of melt ponds on the sea ice, which appear radiometrically similar to open water and can be conflated with open water (Ulaby et al., 1986; Kern et al., 2016), hampering the ability of PM algorithms to ascertain the true sea ice coverage.

Local errors in PM-SIC are observed to have a compensating effect when integrated over the Arctic or Antarctic, and hence the impact of algorithmic uncertainty or bias on estimates of total sea ice coverage is estimated to be less than 1%, even in summer (Notz, 2015; Meier and Stewart, 2019; Kern et al., 2020). Still, no independent alternative to PM exists for measuring SIC from local to global scales. Thus it is not clear whether biases exist in PM-SIC algorithms that go beyond normally-distributed uncertainties, which might affect climate process understanding, forecast model data assimilation, and future projections.

Here we present a gridded SIC product, the linear ice fraction (LIF), developed using NASA's ICESat-2 laser altimeter (IS2). IS2 is a photon-counting laser altimeter with 0.7 m along-track sampling, a 10-meter footprint, and high skill in differentiating sea ice and open water in non-summer months (Farrell et al., 2020; Kwok et al., 2020, 2021). Unlike radar altimeters, IS2 is not susceptible to "snagging" by leads or melt ponds. IS2 can resolve Arctic leads at the sub-meter scale (Petty et al., 2021; Kwok et al., 2021), especially in winter, and has shown a limited ability to identify melt ponds atop Arctic sea ice in summer (Farrell et al., 2020; Tilling et al., 2020). Importantly, IS2 does not rely on the PM signature of sea ice and has independent uncertainties from PM-SIC. LIF is derived from one-dimensional measurements of the sea ice surface with low return periods compared to PM satellites. Its potential use in improving global, two-dimensional measurements like SIC has to be carefully assessed.

We first explore errors and uncertainty in PM-SIC measurements using a set of more than 27,000 classified images from NASA's Operation IceBridge Digital Mapping System (Buckley et al., 2020) in Sec. 2. Using a series of different high-resolution imagery, we show that LIF derived from a single ICESat-2 pass is at least as skilled at PM products at reconstructing local SIC for SIC near 100%. Using an IS2 emulator, we derive bounds on the number of IS2 passes required to guarantee that local errors in a gridded IS2 product are below expected uncertainties in PM-SIC in Sec. 2.3. We then examine global differences between monthly IS2 LIF and six commonly-used PM-SIC products at different resolutions in Sec. 3.

## 2 Comparing Sea Ice Concentration Products to Operation Icebridge Imagery

Operation IceBridge was a multi-year observational campaign to bridge the gap between the ICESat and ICESat-2 satellite operational periods. IceBridge flights captured along-track optical imagery of the sea ice surface. Here we examine a set of more than 27,000 geolocated and orthorectified images taken in April 6, 2018 (pre-surface-melt), and during the July Arctic campaigns in 2016 and 2017 (during surface melt) (Dominguez, 2010). We chose the specific April 2018 flight, the "Laxon



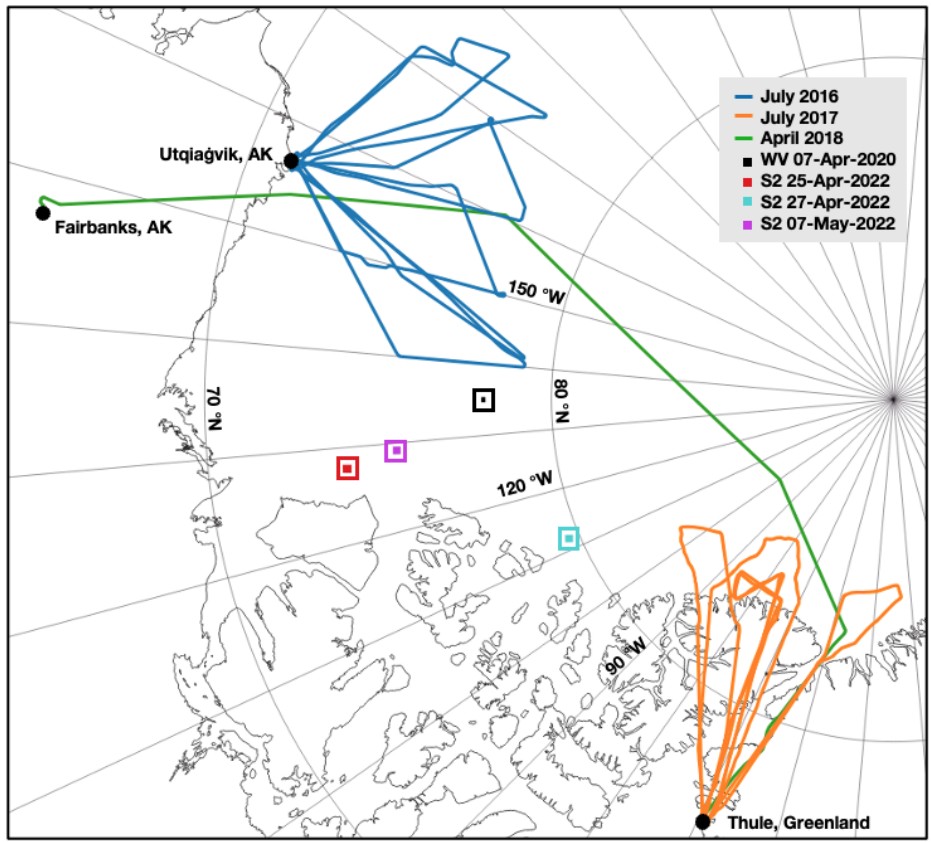

**Figure 1. Imagery Location.** Operation IceBridge flight lines for the July 2016 (blue) and 2017 (orange) summer Arctic sea ice campaigns, and the April 6, 2018 flight line (green). The footprints (boxed) of the WorldView and Sentinel-2 images used for validation of the ICESat-2 LIF: April 7, 2020 (black), April 25, 2022 (red), April 25, 2022 (cyan), May 7, 2022 (purple).

Line", because it samples both multiyear ice and first year ice as the aircraft transits from Greenland to Alaska (Figure 1). The DMS imagery has 0.1 m resolution and are approximately 400 m x 600 m. Each image is then processed according to the classification scheme of (Buckley et al., 2020) (hereafter B20), which segments the sea ice surface into ice, open water, and
melt pond categories. Classified imagery was then visually validated.

The B20-classified OIB scene is compared to local SIC evaluated using six commonly used daily gridded PM-SIC products that use two separate passive microwave satellite platforms. The first four utilize brightness temperatures from the Special Sensor Microwave - Imager/Sounders onboard US Defense Meteorological Satellite Program flight units 16-18. They are (1) the NASATeam (NT) (Comiso, 1986) and (2) Bootstrap (BS) algorithms (Comiso, 2000), the (3) NSIDC Climate Data Record
(CDR), which is equal to the maximum of the Bootstrap and NASATeam algorithms (Meier et al., 2021), and (4) the OSISAF Global Sea Ice Concentration climate data record (OSI450-a, up to 12/31/2020) and interim climate data record (OSI430-a, up to 2023) (Lavergne et al., 2019). We also examine two algorithms that utilize brightness temperature data retrieved



| | July 2016, 2017 | | | | April 2018 | | | |
| | All (n=14769) | | MPF > 25% (n=3578) | | All (n=10702) | | B20 SIC=99%(n=415) | |
| | $\overline{\text{SIC}}$ | $\overline{|\Delta|}$ | $\overline{\text{SIC}}$ | $\overline{|\Delta|}$ | $\overline{\text{SIC}}$ | $\overline{|\Delta|}$ | $\overline{\text{SIC}}$ | $\overline{|\Delta|}$ |
|---|---|---|---|---|---|---|---|---|
| B20 | 85.2 | ∅ | 81.1 | ∅ | 99.8 | ∅ | 96.6 | ∅ |
| CDR | 90.3 | 12.9 | 77.7 | 18.8 | 99.5 | 0.6 | 98.6 | 3.0 |
| NT | 74.6 | 17.0 | 57.6 | 29.9 | 99.4 | 0.7 | 98.3 | 2.9 |
| BS | 90.3 | 12.9 | 77.7 | 18.8 | 99.0 | 1.1 | 98.3 | 3.0 |
| OSI | 79.8 | 14.3 | 62.5 | 26.3 | 98.6 | 1.4 | 98.2 | 2.7 |
| AMSR2-ASI | 97.0 | 14.2 | 98.2 | 18.2 | 98.4 | 1.5 | 97.2 | 2.8 |
| AMSR2-NT | 93.0 | 14.6 | 83.3 | 21.2 | 98.9 | 1.1 | 98.6 | 2.2 |

**Table 1.** Comparison of B20-derived SIC data to common PM satellite algorithms. MPF is melt pond fraction. $\Delta$ is with reference to B20-classified values, and overline is the average over all scenes. Thus $\overline{|\Delta|}$ is the mean absolute difference in SIC values from B20.

from the Advanced Microwave Scanning Radiometer 2 (AMSR2) sensor on board the JAXA GCOM-W satellite, either (5) the NASAteam algorithm (Meier, 2018), or (6) the ASI-ARTIST algorithm Spreen et al. (2008). (1-3,5) are provided on the NSIDC
25km polar stereographic grid. We use OSI450/430 products (4) on the 25km EASE grid and (6) the ASI-ARTIST product on a 6.25km polar stereographic grid. The selection of algorithms and sensors was chosen to best represent the commonly-used PM-SIC products used in the modeling and observational community, and capture the range of PM-SIC observations Kern et al. (2019, 2020).

To compare OIB imagery to any given SIC product, we take the center latitude and longitude of the optical image and
identify that with its nearest grid cell in the native grid of the SIC product, noting that each OIB scene is significantly smaller than a single PM-SIC grid area (0.24 km$^2$ v. 625 km$^2$). We only examine data where all PM-SIC products have a SIC value above 15% to avoid measurements outside the marginal ice zone. In total, this leaves 25,471 unique scenes - 14,769 points of comparison in "summer" and 10,702 in "winter". Comparative statistics for all data are collected in Table 1, which include the mean SIC value ($\overline{\text{SIC}}$), as well as the mean absolute difference between the B20-derived SIC and the PM-derived SIC ($\overline{|\Delta|}$).
For all summer data, average melt pond fraction on summer sea ice was 17.8%, and mean absolute differences are greater than 12.9% for all products. For all winter data, mean absolute differences were less than 1.5% for all PM-SIC products. On average, across all winter scenes, the PM-SIC products underestimated the B20 SIC, which had a mean value of 99.8%.

Sea ice types recorded during OIB are not necessarily representative of all ice types encountered in the Arctic, and were not observed to facilitate an intercomparison of SIC data. To understand potential biases in PM-SIC products, we separate the
B20-clasified OIB data into two categories to explore the impact of melt ponds and leads on PM-SIC retrievals.

We first separate and examine all summer data with heavily ponded ice, which we define as a melt pond fraction (MPF) greater than 25% (the mean MPF of such scenes is 36.6%), of which there are 3578 scenes. Statistics on these data are provided in Table 1, and in Figure 2(a) we show the distribution of SIC values for all such scenes. We note a wide difference



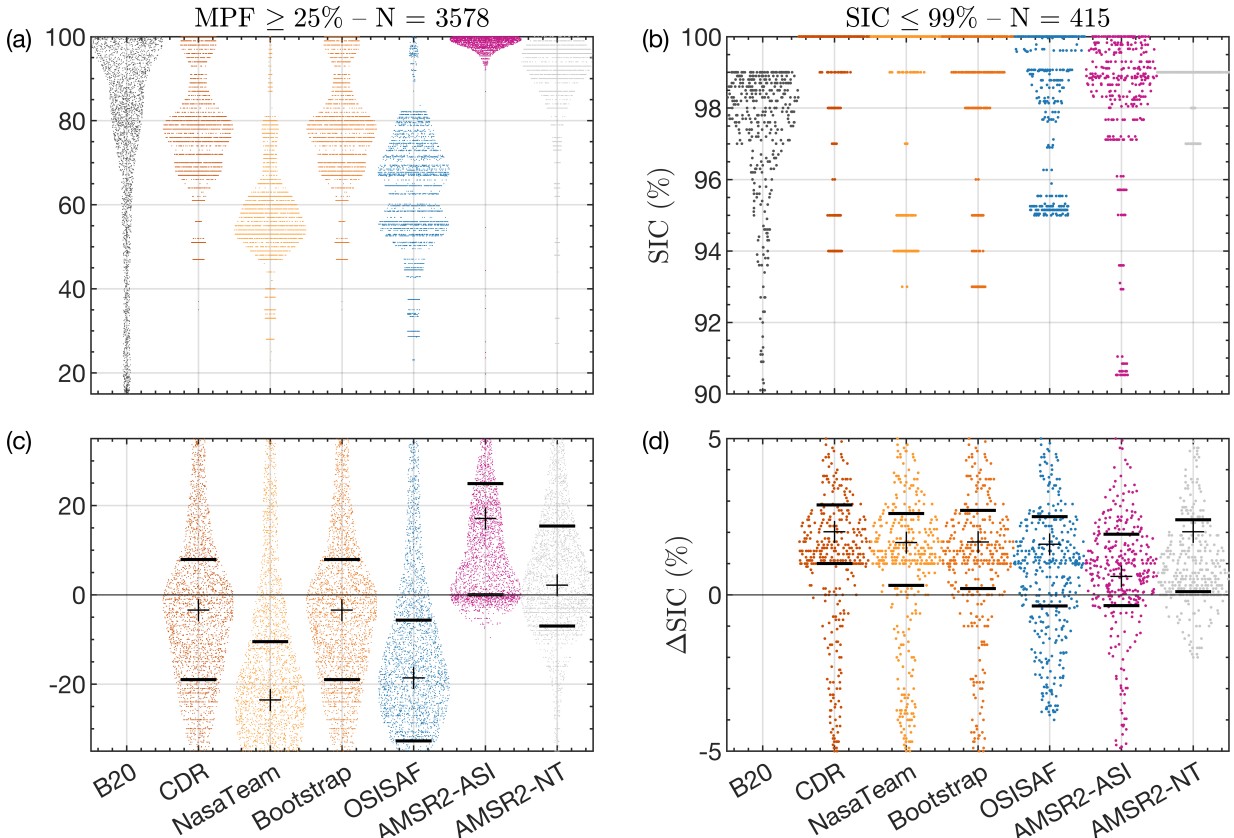

**Figure 2. Differences in PM-SIC retrievals over Operation IceBridge Scenes** (a) Distribution of SIC values for summer scenes with melt pond fraction greater than 25% for each of the SIC products. (c) Difference between PM-SIC and B20 SIC values, $\Delta$. Crosses are median values of $\Delta$. Horizontal lines are the interquartile range of $\Delta$. (b,d) Same as (a,c) but for winter scenes where B20 shows less than 99% SIC.

in distributional shape between the different algorithms. The NSIDC-CDR uses, during this period, the Bootstrap algorithm
alone, and therefore the two match. The worst-performing data are those from the NASATeam algorithm (mean absolute bias
of 29.4%). Both the Bootstrap and OSI products show a doubly-peaked distribution, whereas the two AMSR2-based products
approximate the distributional shape of the B20 data.

The distribution of differences from the B20 data ($\Delta$) are then shown in Figure 2(c), along with median differences (hash)
and interquartile ranges (solid horizontal lines). The SSMI/S products all show a median underestimation of SIC, in contrast
to the AMSR2-based products, which overestimate it. Median average differences in the data are at least 18.2%, although the
distribution of error is highly symmetric in the case of the CDR/Bootstrap products, as well as the AMSR2-NT product. Both
the interquartile ranges of the NASATeam and OSISAF products lie away from zero. Interestingly, the NASATeam algorithm





applied to AMSR2 data performs significantly better at reproducing the B20 data than the NASATeam algorithm applied to SSMI/S data.

We repeat this analysis for OIB scenes in winter as Fig. 2b, but looking only at those where B20 records SIC below 99%, indicating the presence of a lead(s) (415, 4% of winter scenes, and note the lack of B20 points above 99%). Though only a small proportion of wintertime scenes are examined here, the total amount of open water in these 415 scenes accounted for 84% of the total open water area in all winter scenes, and therefore plays a role in total air-sea exchange. When leads are present, all PM data *overestimate* SIC, with mean overestimates ranging from 0.6% for the AMSR2-ASI product to 2.0% for

the Bootstrap product (Table 1). This is largely because most products still report near-100% SIC in these scenes (b). Note that the y-axis limits differ in (b) from (a): as the Bootstrap/NASATeam-based products report their values in unit increments, the distribution of SIC values shown in (b) is highly discrete. With the exception of a similar bimodal peak in the OSISAF data, most of the PM-SIC data is highly concentrated towards 100%. Thus the interquartile range of biases from the CDR, NASATeam, Bootstrap, and AMSR2-NT products (d) does not include zero, and the 25th percentile is -0.36 for the OSISAF

data and -0.35 for the AMSR2-ASI data.

    In general, we find that most PM products have high uncertainty in the presence of ponds with potential sensor-related differences (higher SIC in AMSR2-based products than SSMI/S-based products), in line with other intercomparisons (Kern et al., 2016, 2020). We find that differences between B20-classified scenes and PM-SIC scenes have high spread, with median biases of at least 18.2% (AMSR2-ASI) (Table 1). In winter, while most scenes are almost fully ice-covered, those that have

leads show a consistent overestimation in the PM-SIC data which results in systematic reductions in open water fraction of 18% (AMSR2-ASI) to 42% (NSIDC-CDR). Median biases are at least 2.2% (AMSR2-NT) for open water fractions of 3.4%, which we will consider a core uncertainty to overcome in the development of a complementary ICESat-2-based product below.

    Results presented here have important limitations from uncertainty in surface classification and mismatches between satellite footprints. We discuss the applicability and limitations of this approach in more detail in Sec. 4. Yet because of these consistent

biases, we seek then to understand the applicability of alternative methods for retrieving SIC to reduce, understand, or constrain these uncertainties.

## 2.1   ICESat-2 and the Linear Ice Fraction

ICESat-2 (IS2) is a 6-beam laser altimeter with extremely high precision and skill in retrieving sea ice properties (e.g Kwok et al., 2019b), which may provide an alternative method for retrieving SIC. We use the sea ice height product, known as

ATL07, which compiles sequential along-track "segments" consisting of 150 photons each. Based on the statistical properties of the returned set of 150 photons, as well as their height, each segment can be identified with a surface type (lead, ice, or cloud covered) (Kwok et al., 2019a). IS2 also detects dark or gray ice that might ordinarily be recorded as ocean in passive microwave calculations (Petty et al., 2021). Segments are provided in locations where the local daily NSIDC-CDR sea ice concentration exceeds 15% and average ∼15 m for the strong beam and ∼60 m for the weak beam (Kwok et al., 2019b).





| Date | 07Apr2020 | | 25Apr2022 | | 27Apr22 | | 07May2022 | | |
|---|---|---|---|---|---|---|---|---|---|
| Location | (-135,78) | | (-127.7,73.9) | | (-111.8,79.7) | | (-129.2, 75.4) | | |
| SIC | SIC | $\overline{\Delta}$ | SIC | $\overline{\Delta}$ | SIC | $\Delta$ | SIC | $\Delta$ | $\overline{|\Delta|}$ |
| **Optical** | 97.5 | $\emptyset$ | 97.9 | $\emptyset$ | 98.7 | $\emptyset$ | 90.5 | $\emptyset$ | $\emptyset$ |
| **IS2** Best | 97.5 | 0.0 | 95.7 | -2.1 | 98.7 | 0 | 92.4 | 1.9 | **1.0** |
| ATL07 | 99.3 | 1.8 | 98.3 | 0.5 | 99.8 | 1.1 | 96.6 | 6.1 | **2.4** |
| **PM** ASI | 94.5 | -3.0 | 98.7 | 0.9 | 97.0 | -1.7 | 96.7 | 6.1 | **2.9** |
| OSI | 99.9 | 2.4 | 95.5 | -2.4 | 95.0 | -3.8 | 100.0 | 9.4 | **4.5** |
| CDR | 100.0 | 2.5 | 100.0 | 2.1 | 100.0 | 1.3 | 100.0 | 9.5 | **3.8** |
| NT | 100.0 | 2.5 | 100.0 | 2.1 | 100.0 | 1.3 | 100.0 | 9.5 | **3.8** |
| BT | 99.2 | 1.7 | 100.0 | 2.1 | 100.0 | 1.3 | 100.0 | 9.5 | **3.6** |

**Table 2.** Comparing Linear Ice Fraction and Sea Ice Concentration from ICESat-2, coincident optical Imagery, and Passive Microwave Products.

Over any location, and over any time period, we define the linear ice fraction (LIF):

$$LIF = \frac{\text{length of ice segments}}{\text{length of all surface segments}}, \qquad (1)$$

where we do not include cloud-covered segments. LIF is a one-dimensional analog of the SIC, which can be calculated in a domain either based on a single satellite pass (using all 6 beams), or by compiling many intersecting passes. We pre-process all ATL07 IS2 tracks by removing anomalous segments longer than 200m and eliminating segments that have fewer than two

neighboring segments within 1 km along-track as in Horvat et al. (2020).

## 2.2   Comparison of a single-pass LIF with observations

The LIF product has promise in its ability to improve estimates of SIC, but alone it may not accurately represent a two-dimensional field like SIC. To understand its applicability in single test cases, we examined four high-resolution images co-incident in space and near-coincident in time with IS2 overflights in regions with a high concentration of leads, three from

Sentinel-2 optical imagery and one from the WorldView satellite (shown in Fig. 3). WorldView-2 is a member of the Maxar WorldView Legion with commercial satellites providing high resolution multispectral imagery. The red, green, and blue bands have 1.85-m resolution, higher than the IS2 footprint. The Sentinel-2 (S2) mission consists of a pair of satellites carrying the multispectral instrument (MSI) acquiring data in 13 bands. The red, green, blue, and near-infrared bands (B02, B03, B04, and B08) are at 10-m resolution, a resolution similar to the IS2 footprint (Drusch et al., 2012). We examine 25 km x 25 km areas of

the Sentinel-2 imagery with the ICESat-2 tracks intersecting 25 km of the image. The ICESat-2 tracks transect the 14 km x 17 km WorldView image for  14.2 km. Following (Buckley et al., 2023), we classify the WorldView and Sentinel-2 image pixels into surface types: open water, ice, and other. The other pixels in these scenes are associated with new ice that appears gray in color which, for SIC and LIF calculations, is considered ice.



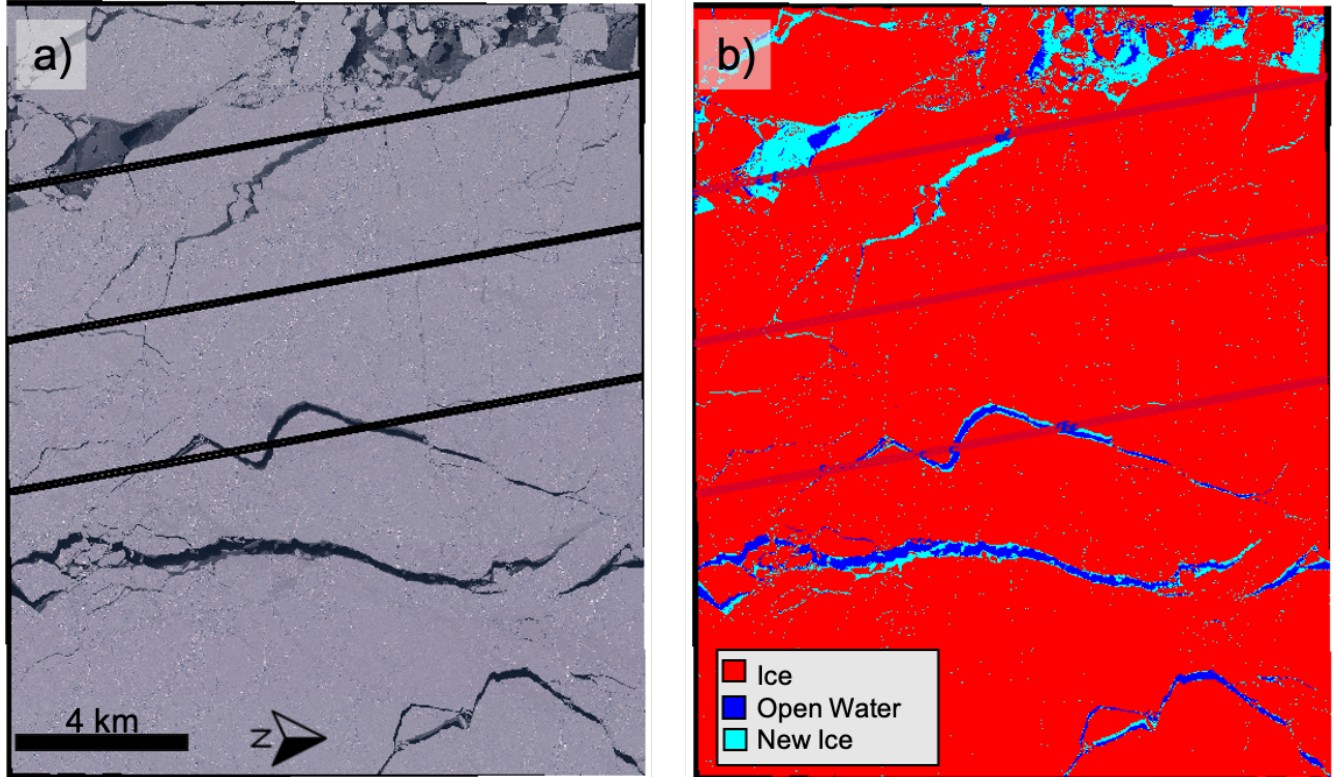

**Figure 3.** (Left) RGB Worldview-2 image taken on April 7, 2020. Straight lines are the overflight of the ICESat-2 laser altimeter. (Right) Classification of the image by the B20 algorithm into open water, new ice, and sea ice.

We calculate two different LIF metrics for each scene. First, we calculate a "best" LIF by extracting the value of the clas-
sified WorldView imagery at the location of each ATL07 segment for all six IS2 beams. Second, we use the values of the ATL07-derived surface type classifications after post-processing to determine the ATL07-derived linear ice fraction (Eq. 1). We compare these against the local values of optically-derived SIC from the imagery and the local PM-derived SIC (bound to the area of the optical imagery) from the five products listed in Table 2. In each case we compute biases relative to the optical imagery SIC. Note that while all observations happen on the same day (Optical vs. IS2 vs. PM), the overflights of satellites are 155 not synchronous, so some biases could emerge because of changes to sea ice over that domain between satellite passes, though we expect these to be small.

In Table 2, we tabulate the biases for all images, and compile the average imagery bias by taking the mean of the absolute bias across the four images. Even for a single pass, the "best" and ATL07-based IS2 LIF outperforms the PM-SIC products, with a mean bias of 1.0% and 2.4%, compared to mean biases of at least 2.9% for the PM products. This is especially notable 160 in the May 7, 2022 image, an area of highly fractured sea ice which is erroneously treated as being completely ice-covered by four of the PM satellites. In all cases, the NSIDC CDR estimates 100% SIC, though the imagery shows between 1.3%





and 9.5% open water fraction. The standard ATL07 product outperforms the PM products, with a median error for ATL07 classification is similar to the best-case error for all PM-SIC retrievals in OIB data (see Table 1), but there remains substantial room for improvement - a further 60% improvement above the ATL07-based LIF is possible, to a "best" bias of just 1.0%, in these imagery. Thus it is clear that improvements in classification could lead to an IS-2 based SIC product that improves substantially upon the error characteristics of PM-SIC data in high-concentration ice regimes.

## 2.3    Error bounds on IS2-SIC from emulation

The analysis summarized in Table 2 provides confidence that IS2 could be used to estimate SIC, although there remains a bias associated with the orientation and location of leads and the IS2 flight path even in the case of a perfect classification. The bias for the four high-resolution images examined above is approximately 1.0%. These scenes were comprised of compact ice with relatively diffuse leads; in more complex scenarios with larger open-water fractions, IS2 passes may not accurately sample the ice surface and therefore LIF may be unable to estimate the scene's SIC.

To understand how well we should expect IS2 to reproduce SIC, we build an LIF emulator. An example of the emulator is shown in Fig. 4, using the April 7, 2020 WorldView-2 image. A binary sea ice mask is intersected randomly with straight lines (a), which we call a "pass", and each pass is identified by the classification as open water or sea ice. The along-track sea ice concentration if calculated for each pass (red crosses, b), which is then compared to the "true" image SIC (B20 classification), resulting in an "SIC error" (c). Each surface is then sequentially intersected by more passes (identified by the "crossing number" in Fig. 4(b,c)). The cumulative estimate of SIC with each sequential crossing is given as a blue line in (b), with the measurement made at each crossing given as a red cross. The cumulative error decreases in time as more of the surface is sampled, although we see that the emulated IS-2 passes typically outperform the PM products even for a single pass, as was examined in Table 2.

To understand the statistics of retrieving SIC over multiple passes, we iterate this emulation procedure over a subset of 30 randomly-selected OIB images. We repeatedly emulate SIC retrieval as in Fig. 4b for each image, forming an ensemble estimate of the error statistics from a series of trials. For the 30 different OIB images, we plot the median SIC error per crossing number as a gray line, with gray shading the interquartile difference across all images and iterations. Median errors drop below ±2.2% (red dashed lines), the minimum wintertime bias we observed for OIB imagery in Table 1, after just two passes, with the interquartile range doing so after eight passes (vertical dashed line). We will use this number as a baseline when considering unsupervised retrievals of LIF from IS2.

There are important caveats in applying this approach to understanding IS2-related bias. First, the domain of interest is smaller than the distance between strong beams of 3km for IS2. Therefore, we consider just a single beam, which permits us to analyze LIF derived on the scale of a single optical image. The error bounds here may be improved when using the full six-beam IS2 overflights, and we do so in Sec. 3. Second, the error bounds derived here utilize a "best" ATL07 classification, though in practice the identification of open water and sea ice is imprecise, which can introduce enhanced bias, as indicated in Table 2. Although in the cases examined here, this increased bias is less than that from the PM-SIC sensors. Finally, the underlying ice is evolving in time between crossings, not static as is assumed here. The use of the ensemble estimates across different OIB images helps to understand the variability in error between different and evolving sea ice surfaces.



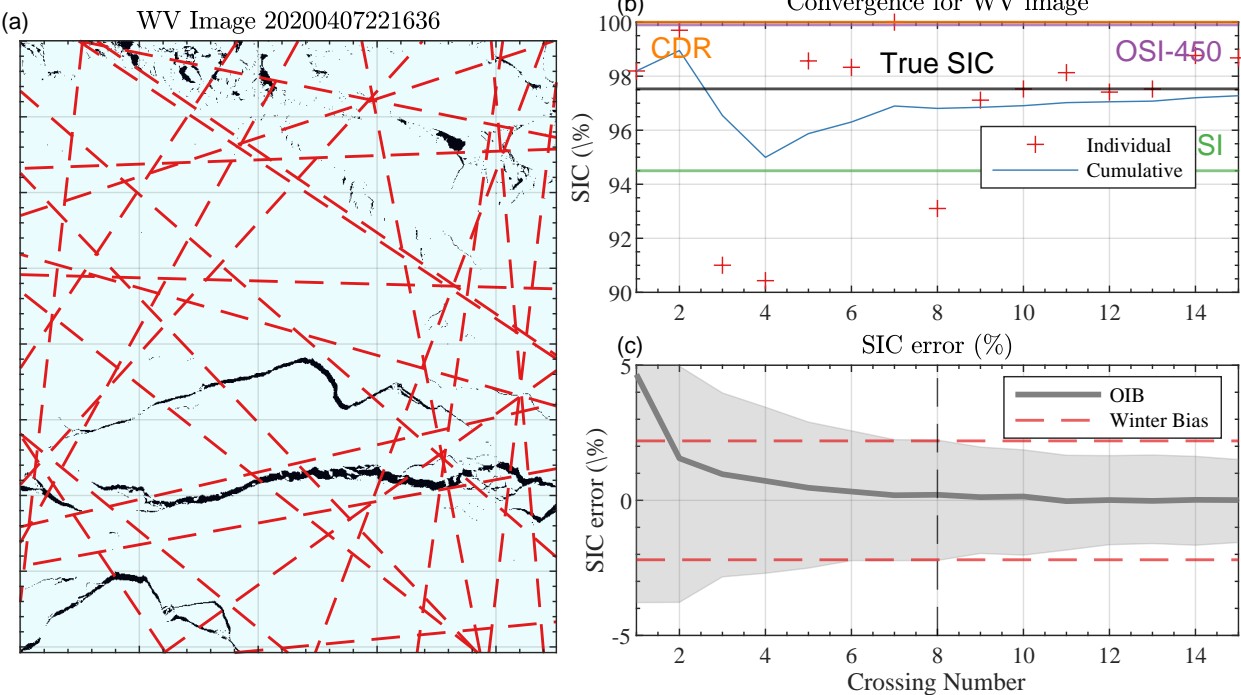

**Figure 4. Error statistics for an IS2-based emulator** (a) Ice-ocean binary mask using the B20 algorithm applied to the April 7, 2020 WorldView image with example emulated IS-2 tracks overlaid on it. (b) Estimated SIC for the WV image, comparing the true value (black horizontal line) to three PM products. Individual emulator trials are given as crosses, and the cumulative SIC from the IS2 emulator is in blue. (c). Statistics of SIC error for all OIB images as a function of crossing number, and a series of synthetically-generated sea ice surfaces. Solid lines are the average at intersection number, with the variance across 200 unique estimates given by the shading. Approximate error for winter images is given as the two dashed lines.

Overall, this emulator helps confirm the applicability of these single-pass methods (akin to its use in deriving the floe size distribution (Rothrock and Thorndike, 1984; Tilling et al., 2019; Horvat et al., 2019)) and for establishing the decay of error bounds with increasing number of overflights. In consideration of the emulator and associated errors, and under the assumption that they are uncorrelated in time, here when forming a global IS2 product, we will only consider regions that include at least 200   5 overflights (a total of 15 strong-beam crossings and 30 strong-and-weak-beam crossings).

## 3   A Global IS2-based LIF Product

Above, we showed that IS2 can be used to reconstruct the fraction of ice coverage with high precision, and that expected errors in IS2-based SIC retrievals asymptote quickly, with important caveats. We next seek to use this information to build an



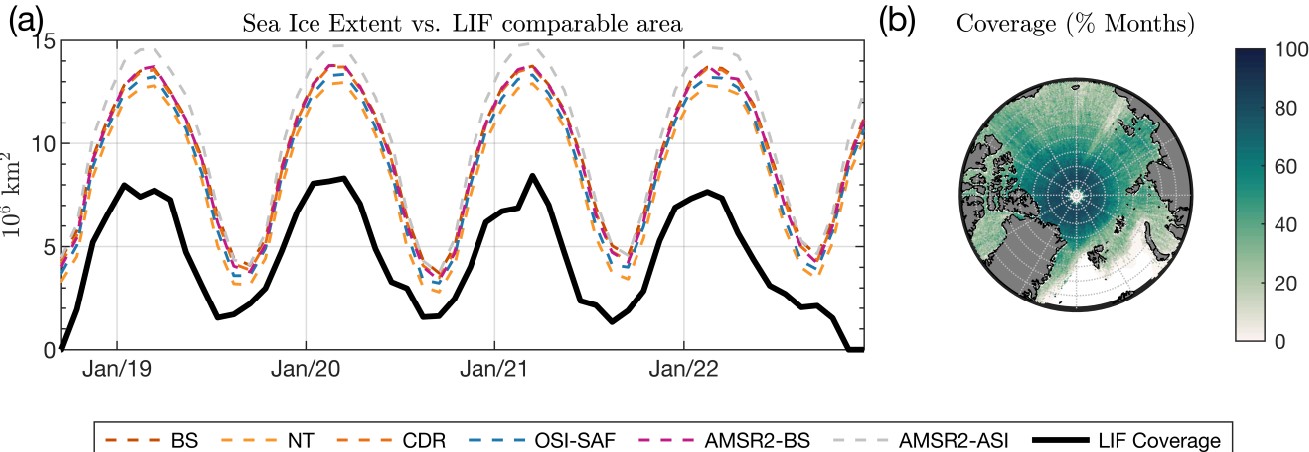

**Figure 5.** Comparison of coverage of IS2 LIF data to commonly-used PM-SIC products. (a) Arctic sea ice extent of 6 PM-SIC products (dashed lines) compared to the area well-sampled by IS2 (solid line) from October 2018-December 2022. (b) Percentage of months from October 2018-December 2022 where PM-SIC record sea ice and IS2 tracks are sufficiently dense.

IS2-based SIC product in the Arctic and Southern Oceans. We provide the Antarctic data in this study, but we will not analyze it here.

### 3.1 Tradeoffs in temporal sampling

The intermittency of IS2 overflights does not allow for high-frequency returns as are possible with daily PM measurements, which contributes to the orientation errors discussed in Sec. 2.3. To integrate many crossings at the scale of a typical PM-SIC measurement, we therefore create a monthly LIF product, on the 25km polar stereographic grid, the same resolution of target PM-SIC products. When integrating temporally distant IS2 overflights, sea ice's time evolution must be considered, as IS2 will not be able to pick up on daily variations that PM-SIC might and this could introduce spurious differences in the monthly-averaged IS2-based product and PM-SIC products. Thus at each grid point, we define define an "ICESat-2 intermittent" PM-SIC, $\tilde{c}$, equal to the average sea ice concentration observed in the NSIDC-CDR SIC record on days that ICESat-2 passed overhead. We then define an "intermittency bias", $B_I$ from the monthly-average CDR SIC, $\bar{c}$, as,

$$B_I = \tilde{c} - \bar{c}. \tag{2}$$

The value of $B_I$ is a measure of how different the PM-SIC product would be if sampled only when IS2 flew overhead from the true PM-SIC monthly mean, and estimates the bias introduced by IS-2's intermittent sampling. Conscious of the impact of examining a time-evolving SIC surface, we ignore all grid cell where $\|B_I\|$ exceeds 0.5%. This reduces the number of grid cells over which we develop an LIF product by up to 75% compared to the PM-SIC products.



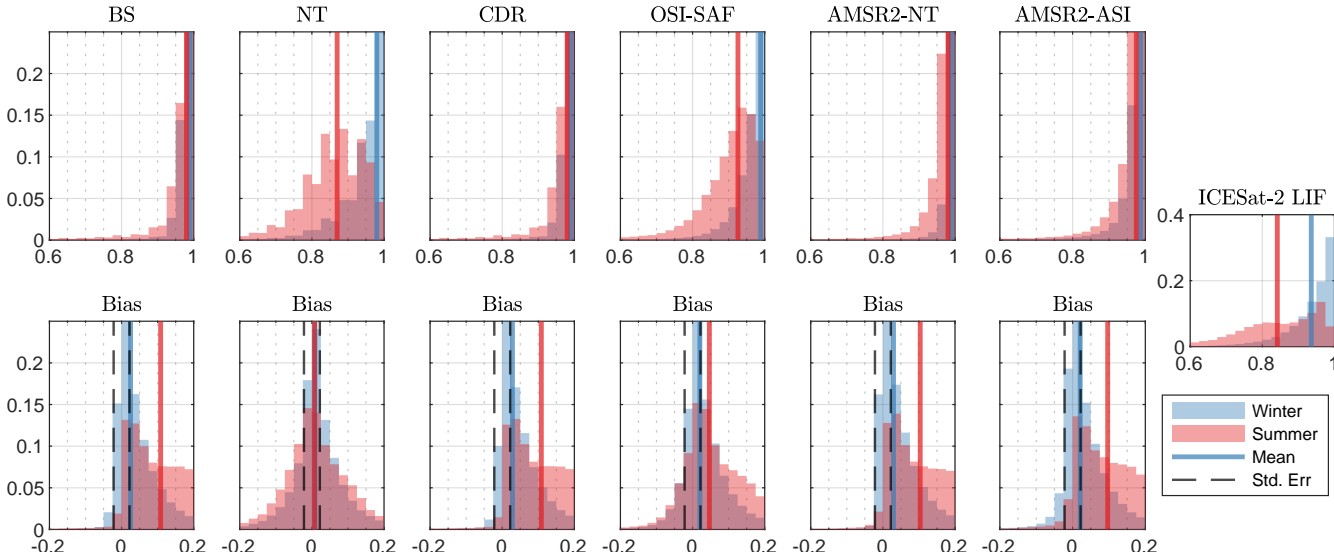

**Figure 6.** Histograms of PM-SIC for 6 products (top row) and difference from IS2 LIF (bottom row), with summary statistics provided in Table 3. Red colors are summer months, blue are winter months. Vertical lines are respective median SIC and $\Delta$ values. Vertical dashed lines are $\pm 2.2\%$, the minimum wintertime bias we observed for OIB imagery in Table 1.

We couple this reduction with the requirement of at least 5 IS2 crossings discussed in Sec. 2.3. We also require that all SIC estimates indicate above 15% SIC at a given location to make a comparison. In Figure 5(a) we show the total sea ice extent for all PM products (dashed lines), compared to the area over which we make an IS2-PM comparison (solid line).

We show in (b) the fraction of months during the IS2 operational period (here from October 2018-December 2022) where there is sea ice recorded by PM-SIC and sufficient IS2 tracks. Because of the higher track density near the pole, this is nearly all months above 80°N, with a reduction in LIF coverage farther from the pole. In total, in any given month the area with compatible IS2 coverage for comparison with PM-SIC is between 25% and 65% (Fig. 5a) of total sea ice extent. The lowest percentage coverage is in late summer months when the rapid decline of sea ice introduce larger intermittency biases.

We next build the gridded LIF product by accumulating all filtered ATL07 segments into a 25km polar stereographic grid over a given month. For building the LIF, when examining open water segments, we select all that are not classified as ice using the ATL07 SEG_SURF_TYPE flag. This flag also discriminates between "specular" and "dark" leads, and we additionally provide an LIF product ($LIF\_s$) which only counts those specular returns as open water as it may be of interest given the radiometrically different properties. For a global comparison with PM-SIC, in Fig.5 we display the histogram of SIC values (top row) and histogram of differences from LIF (bottom row) for all data for each PM-SIC product. Data is segmented into "summer" data from June to September (red), and "winter" data from October to May (blue), with relevant median values as vertical lines. The minimum winter lead uncertainty from OIB imagery is shown as vertical dashed lines for $\Delta$ plots.

Statistics derived from these distributions is given in Table 3, along with interquartile ranges and median differences from LIF (shown using the symbol $\tilde{\Delta}$. In total, we have approximately 54,000 "summer" comparison points covering 36 million



| Period | "Summer" (Jun-Sep) | | "Winter" (Oct-May) | |
|---|---|---|---|---|
| Number | $54 \times 10^3$, | | $301 \times 10^3$, | |
| Area | $35.8 \times 10^6$ km$^2$ | | $195.1 \times 10^6$ km$^2$ | |
| | $\overline{SIC}$ | $\tilde{\Delta}$ – (25%,75%) | $\overline{SIC}$ | $\tilde{\Delta}$ – (25%,75%) |
| LIF | 84.1% | $\emptyset$ | 93.6 | $\emptyset$ |
| LIF (specular) | 91.2% | 5.3% (1.8,10.7) | 98.7 | 2.9% (0.3,7.3) |
| Bootstrap | 96.5% | 10.8 (4.0, 19.2) | 98.2% | 2.5% (0.2,7.0) |
| NASATeam | 85.4% | 0.7% (-3.9,6.6) | 95.4% | 0.8% (-1.6, 5.0) |
| NSIDC-CDR | 96.5 | 10.8% (4.2,19.2) | 98.5% | 2.8% (0.4, 7.2) |
| OSI-SAF | 90.2% | 4.7% (0.6,11.9) | 97.1 | 2.0% (-0.0,6.2) |
| AMSR2-NT | 96.3% | 10.4% (3.9,18.9) | 98.6% | 3.0% (0.5,7.4) |
| AMSR2-ASI | 95.2% | 9.7% (3.3,17.7) | 97.3% | 2.1% (-0.2,6.5) |

(Product label spans the Product rows.)

**Table 3.** Comparison of "summer" (Jun-Sep) and "winter" (all other months) statistics of IS2 global LIF product and the set of 6 examined PM-SIC products. $\Delta$ values are differences from standard LIF product. $\tilde{\Delta}$ is median difference. Values in parentheses is the interquartile range of $\Delta$ (25%-75% intervals).

km$^2$, and 301,000 "winter" comparison points covering 195 million km$^2$. We see generally that PM-SIC products report a higher ice fraction than the LIF. Wintertime biases are similar to that found in OIB data as well as in classified optical data, with overestimations of 1-3% for sea ice that is on average 93.6% ice-covered according to LIF. Specular returns showed similar overestimation biases in winter to the PM-SIC products, pointing to the possibility that the non-specular returns are open water areas that are not sampled in passive microwave signatures. Because of the absence of surface meltwater, these non-specular returns are less likely to be contaminated by misclassification error in winter, and may be the result of a true PM-SIC overestimation bias.

Summer $\Delta$ values have a more variable spread, and generally a positively-skewed distribution of summer values which is opposite in sign to our findings in OIB data in Fig. 2, and Table 1. The LIF and NASATeam PM-SIC algorithms are most similar, with the smallest mean difference and an interquartile range that includes zero. As NASATeam data were the worst-performing when compared to the highly-ponded OIB imagery, we suggest this points to known challenges in ATL07 segment type classification, which at this time is unable to readily distinguish between leads and ponds (Kwok et al., 2019a; Tilling et al., 2020; Buckley et al., 2023; Herzfeld et al., 2023) (see Discussion), and this similarity may suggest melt ponds are likely counted as open water in the LIF calculation. Using only summer leads classified as specular resulted in similar distributions of LIF values to those from the PM-SIC products, though errors in the retrieval of summer sea ice properties from ICESat-2 should continue to be the object of future study, to establish whether PM-SIC products are overestimating SIC in the melt season on average or not.



## 4   Conclusions

In this study, we evaluated the skill of commonly-used PM-SIC algorithms to represent local sea ice concentration, compared to high-resolution optical imagery from Operation IceBridge, optical satellite sensors, and an estimate of sea ice concentration from individual passes of the ICESat-2 laser altimeter. We showed that, in general, all PM-SIC measurements had positive biases in winter conditions over compact sea ice, consistent with existing literature Kern et al. (2019). For near-100% sea ice, the ICESat-2 altimeter can produce comparable estimates of SIC even for a single overflight of a sea-ice-covered area, and with improvements to the ATL07 surface classification scheme has room to reduce open water biases significantly.

In summer, we found a wide spread of differences between algorithms that can be differentiated by the remote sensing platform. SSMI/S-based algorithms tended to underestimate SIC in heavily ponded scenes, whereas AMSR2-based products overestimated SIC compared to B20-classified OIB imagery. While a precise algorithmic and sensor comparison is not within the scope of this study, it invites future work to understand why, on these subsets of data, there is such a systematic difference.

We then explored the creation of a new sea ice concentration product independent from the passive microwave signature of sea ice, the ICESat-2 linear ice fraction (LIF), which we provide as a global, monthly product covering 25-65% of the Arctic sea ice zone. This data product is available through December 2022 (see Data Availability). Because of the available comparative data from Operation IceBridge, we only included Arctic comparisons in this work, though the data product is available in both hemispheres. When compared to PM-SIC measurements over regions with a high concentration of IS2 tracks, the LIF product showed significant and negative differences between IS2-LIF and PM-SIC data in both summer and winter.

In months from October-May ("winter"), we found that the offset between LIF data and PM-SIC product data was of the same order of the bias between the OIB optically classified imagery and PM-SIC data we found in Section 2. Because of this consistency, we suggest that this captures an overestimation bias in the PM-SIC products, and this offset is not from misclassification error in the ATL07 product. Examining an LIF product using only ATL07 "specular" returns resulted in estimates of SIC more similar to PM-SIC, which points to non-specular open water returns as a possible source of the PM-SIC overestimation bias.

In summer, lower LIF values compared to PM-SIC contrasted with expectations for heavily ponded sea ice in OIB imagery, where we found SSMI/S-based PM-SIC products underestimated SIC. As in winter, the classification of specular returns alone as open water resulted in a similar distribution of SIC values as PM-based products. We found that the LIF product was most comparable to the NASATeam algorithm applied to SSMI/S data in these months, a data product which had the largest biases compared to "ground-truth" OIB imagery. Known issues of misclassification by IS2 of melt ponds as open water is the primary source of the negative offset in the LIF product, and find the LIF product is likely not yet capable of enhancing PM-SIC products in these months. Further work on the classification of ponded surfaces is needed before using a summer LIF-based SIC product, and we suggest that the best option for melt season LIF measurements may be the use of specular returns only, which has similar statistics to PM-SIC data.

As it illuminates biases, particularly in compact sea ice in winter, LIF offers an independent and unique opportunity to enhance estimates of sea ice concentration. Underestimations of SIC in the wintertime Arctic may be small: but these differences



correspond to large increases in open water fraction, which can drive ocean and atmospheric variability. Climate models that are tuned to reproduce SIA from PM satellites, or that assimilate PM-SIC for forecasts, may underestimate the magnitude of this air-sea exchange. We have attempted to provide validative data for LIF by using high-resolution optical imagery and an emulation tool. It will be necessary to enrich this LIF data with more constraints to ascertain the year-round and repeat skill of LIF and its potential for developing a new SIC data product on shorter timescales. ICESat-2 offers a high-resolution and repeatable opportunity to provide improved PM-SIC measurements and greater understanding of overall sea ice variability in the polar seas.

*Data availability.* The monthly LIF product, and statistics from operation icebridge and worldview imagery will be provided as a Zenodo repository upon paper acceptance.

*Author contributions.* CH conceived of the LIF product, performed the IS2 analysis, and wrote the paper. EB developed the classification algorithms,provided OIB data and analysis of PM bias. MS and PY developed the IS2 emulator and tested its applicability on data. All authors consulted on the scientific approach and content and EB and MMW contributed to writing the paper.

*Competing interests.* The authors declare no competing interests.

*Acknowledgements.* C.H., M.S., and P.Y. were supported by Schmidt Futures—a philanthropic initiative that seeks to improve societal outcomes through the development of emerging science and technologies. C.H. acknowledges support from NASA 80NSSC20K0959, NASA 80NSSC23K0935, and NSF 2146889. EB and MMW were supported by the Office of Naval Research (ONR) Arctic Program (N00014-20-1-2753, N00014-22-1-2741, and N00014-22-1-2722) and the ONR Multidisciplinary University Research Initiatives Program (N00014-23-1-2014).





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
