# Peer review of "Linear Ice Fraction: Sea Ice Concentration Estimates from the ICESat-2 Laser Altimeter"

_EGUsphere, 2023_

## Author Comment (AC1)

Dear Dr. Howell and reviewers,

Thank you for these extensive and constructive comments. In light of the work we desire to implement to improve the manuscript, we found we could not do so in a single submission, as it would (as pointed out by the reviewers) contain too many things addressed in insufficient detail. Thus, we split the manuscript and will resubmit it in two parts.

Part 1 will focus on enhanced ground-truthing of the IS-2 LIF for single passes using more detailed visible and PM data. It will discuss existing PM SIC products, ICESat-2's dark/grey leads, and more coincident data from Worldview and OIB data.

Part 2 will focus on developing gridded data from IS2 along-track LIF, including a more robust discussion of the error properties of emulation and grinding versus those of PM.

In general, we hope to improve the discussion of validation, algorithmic error, and evaluation - and to address all reviewer comments in our revised articles, although we understand that the review comments cannot be included in these new versions. We hope, then, that you might be available to review our new efforts and reference your comments at that point.

Below, we include our review response document so that future reviewers, editors, and interested scientists can see the implemented changes at this stage.

Best,

Christopher Horvat and Ellen Buckley.

**Reviewer 1:**

In this study, the authors utilize data from ICESat-2 to develop a new observational estimate of sea ice concentration, aiming to validate existing estimates derived from the widely-used passive microwave sensing method. This new 'linear' concentration estimate is initially validated using imagery from the Operation IceBridge airborne campaign.

General comments: The objectives of this study appear quite sensible to me, and it is evident that considerable effort has been invested in the analysis. However, I have significant reservations about the methodology, which cause me some concern, as i detail here:

1. The study attempts to cover a broad range of objectives in a single paper: a. Validating ice concentration estimates from various passive microwave data products using Operation IceBridge imagery during winter and the more challenging summer period. b. Validating ice concentration estimates from ICESat-2 with Operation Ice-Bridge imagery. c. Employing ICESat-2 to generate new ice concentration estimates and to understand linear profiling sampling errors as a function of beam crossings. d. Creating a new global gridded Sea Ice Concentration (SIC) product from ICESat-2 data.

Indeed, the large number of objectives hampered our ability to conclusively describe the methodology of each, and so we split the manuscript to two parts - one handling (a,b) and the other (c,d).

My confidence in the conclusions drawn and the progression from step a to step d is not particularly strong. The validation efforts are commendable but seem somewhat limited. It appears the approach was driven by the desire to demonstrate clear PM biases based on the OIB data, thereby necessitating ICESat-2 data, but this was not always convincingly presented. I wonder if a more effective strategy might involve using more high-resolution imagery for validation and potential bias correction of PM data, rather than introducing the complexities of ICESat-2 data? The inclusion and description of a monthly gridded ICESat-2 SIC product felt overly ambitious, particularly considering the later admission that ICESat-2 struggles to distinguish between melt ponds and leads.

We have added more high-resolution imagery and comparative analysis, and improved our averaging approach for PM/IS-2/Worldview comparisons in part 1.

2. The results in Table 2, showing the difference between IS-2 and imagery compared to IS-2 and PM data (2.4 vs 2.9-4.5), are interpreted by you more optimistically than I would. Given the significantly higher resolution of IS-2 data compared to PM, these results are somewhat disappointing. The much better performance for 'Best' (1.0) is intriguing and may indicate issues with IS-2 data that need addressing, considering other uncertainties (like sampling) and potential unknowns in using altimetry data for this purpose.

We have altered the interpretation from being overly optimistic, but we remain so as it is not entirely given that IS-2 should produce even comparable statistics to heavily-validated PM products. We include significant discussion of the error characteristics in Part II. Additionally, we add further high-resolution scenes in Part I to improve our Table 2 error statistics.

3. Regarding PM products overestimating concentration in winter, I believe it's important to distinguish between actual open water and re-frozen leads, and how these are represented by different sensors. The effectiveness of ICESat-2 in classifying surfaces under various conditions was not thoroughly explained, and the discussion on specular versus dark leads was confusing. It's unclear how many leads are classified as dark and whether any of these were included in the OIB or S2/WV validation.

We added further discussion of the distinction between specular and dark leads in the text, with references to other studies investigating this challenge. The interpretation of lead type from IS-2 is not the goal of this manuscript, but it is certainly a heavily-focused-on problem in the IS-2 community.

4. The explanation of PM data spanning pages 3 and 4 was difficult to follow. A more focused and detailed discussion on PM data, its production, and a comprehensive account of its uncertainties and biases would have enhanced the paper.

We added such a discussion to the new manuscript Part 1.

5. I was surprised that the imagery comparisons did not include any potential drift correction or optimal correlation approaches. Even a minor shift (the size of a lead) could result in misclassification in a given scene. This issue seemed to be dismissed too easily.

We improve the discussion of potential errors from drift in both Parts, but note that these errors are bounded by the "perfect" IS-2 classification, which yields the small drift + misclassification error.

Specific Comments:

L40: Sub-meter scale? At L129, you discuss segment lengths of 15 m and 60 m. Could you clarify this? I am somewhat perplexed about the resolution and what can realistically be resolved with ICESat-2.

We expand the discussion of IS-2 resolution.

L40: Could you elaborate on what you mean by 'snagging' in this context? I believe it still plays a role, does it not?

Snagging is a challenge facing radar altimeters - we explain.

L58: Would you mind adding a reference for DMS imagery and providing more details? I am not overly familiar with this data.

The reference is already included when the product is first introduced (Dominguez 2010), and we re-reference here.

L60: The comparison between OIB and PM is not entirely clear to me. Do you conduct this comparison for each OIB scene, even when multiple scenes span a single PM scene? This approach seems somewhat unconventional. Also, when focusing solely on the 'lead' scenes, wouldn't the locations be effectively identical?

We have improved the averaging of OIB scenes vs. PM scenes to address this comment.

Figure 2: This figure is somewhat challenging to interpret. Have you considered presenting it as a violin plot for better clarity?

These are violin plots, more specifics here would help.

L127: Regarding the use of dark leads, I was under the impression that these are no longer employed in determining SSH in ICESat-2 sea ice products. Could you clarify?

We discuss - this is only for the sea ice thickness products which sensitively depend on the sea surface height location.

L129: The sentence here is confusing. How does segment length correlate with resolution?

We fix in the updated discussion.

Figure 3: If I understand correctly, it is quite surprising that in several instances, the difference between the SIC from the 'best' profile and the imagery SIC is zero. Does this imply that the linear beams are accurately capturing the entire scene? For instance, in Figure 3, the IS2best mean concentration in that scene is 97.5, identical to the mean concentration from the image.

We believe this to just simply be an artifact or luck, along with the rounding to a single decimal place to match the precision of PM data.

L140: For S-2/WV, it appears the scenes are significantly larger than those of OIB, thus offering comparable resolutions. Could you possibly illustrate the PM box in the regions shown in Figure 3?

We add in the caption that this is of similar size to the PM sensor.

Figure 4: I found this figure quite complex to decipher. I would strongly recommend

enhancing the caption and layout for better readability. The transition between OIB and WV, as well as the mixture of legends and plain text in the plots (particularly the PM data), was somewhat disorienting.

Indeed, we split this figure in Part 2 into two figures that are better described in the text.

Figure 4: Why does the Crossing Number start at 0? Should it not be 1?

The missing "1" in the x-axis makes it appear to start at 0! We add a 1 now.

Figure 4: Regarding the initialization of the line in Figure 4b, it's somewhat unexpected how accurate the first one appears, especially considering that it seems plausible to find a crossing without a lead. Could you explain this?

In our experience, the presence of leads/cracks makes it quite challenging to *not* find leads at this resolution, but you notice that the envelope of initial crossings is more broadly biased in Fig. 4. We discuss further and add more examples of "worse-performing" initial crossings.

Figure 4: So, the cumulative SIC converges towards an accurate value as you continue averaging, but doesn't this also apply to the PM datasets if averaged? In practice, would you not be taking just 1 or 3 crossings and using that raw value? In such a case, might the error be similar to that of the PM data, or am I misunderstanding something?

We operate under the assumption that IS-2 errors are random errors, which we showed in both emulation and in the high-resolution imagery. PM errors have a systematic *bias* that will not be improved with average. We explain further in Part II.

Figure 4: The explanation of 'winter bias' is a bit unclear to me.

This has been rewritten.

Figure: Why opt for using the median error?

We prefer non-parameteric statistics here since we do not know the underlying error distribution.

L201: What exactly do you mean by 'high precision'?

We explain further.

L215: Are you suggesting that if the concentration on the day IS-2 passes over a grid-cell matches the monthly mean concentration, you assume there are no sampling biases in that grid-cell at that time? This seems like an unusual approach, especially if you believe there are biases in the SIC data that need addressing.

We only compare local-in-time statistics in both. We explain further.

Figure 5: Referencing my main points above, I remain unconvinced about the value of this, given my significant concerns regarding the interpretation of the 'validation' efforts.

We agree but would like to provide this as a target - and one that has promise. We discuss further.

L242-244: "Due to the absence of surface meltwater, these non-specular returns are less likely to be contaminated by misclassification error in winter, potentially indicating a true PM-SIC overestimation bias." Could you elaborate on this? What about the influence of clouds?

Clouds are excluded in the analysis of IS2 photons as in the segments we examine. We add a short discussion in Parts I/II.

**Reviewer 2:** In this manuscript the authors develop a Linear Ice Fraction (LIF) product from the ICESat-2 satellite ATL07 sea ice height product. The LIF product is designed to be independent of the Passive Microwave (PM) products currently available, and preliminary analysis shows comparable results to those from PM with the benefit of similar-or-better error qualities. It's really exciting to see people thinking outside the box for ICESat-2 and developing new and much-needed applications for the data. However, I have some significant issues with the methodology that should be addressed before publication.

Those significant issues are outlined below, followed by some Specific and Technical Comments.

The comparison of OIB imagery to various SIC products (Section 2) was unconvincing

I appreciate the author's wanted to justify the need for a non-PM SIC product, especially during summer months. But the vastly different scales of the comparison limit its effectiveness. If this section is to remain in the manuscript, more analysis is needed on the spatial variability of SIC on the two different scales, to confirm that such a comparison is meaningful.

We address this issue by taking averages over OIB images to the scale of PM cells in the revised work.

Emulator design

The emulator is developed by randomly intersecting imagery with straight lines, at various intersection angles. This is not representative of how ICESat-2 beam geometry actually samples the sea ice surface. The angle of crossovers is surely a key consideration due to lead geometry – most regions have a typical lead orientation (e.g. Brohan and Kaleschke 2014), meaning leads will be somewhat consistently aligned along or across a given ICESat-2 track in a given region, rather than the random alignment depicted by the emulator. The first paragraph of Section 3.1 refers to "the orientation errors discussed in Sec. 2.3." but I can only see a brief mention of biases associated with lead orientation in the first sentence of Section 2.3. And nothing was discussed regarding the orientation of crossovers and how representative (or not) this makes the emulator.

Part II now includes a greater discussion of the error characteristics of IS-2 intersections with imagery using actual IS-2 tracks over an image.

Not fully considering the implications of including dark leads from ICESat-2 in LIF calculation

The assumption that new/gray ice is considered ice for LIF calculations (L147-148) is too simplistic. Although the author's reference the Petty et al. 2021 paper, they fail to consider that new/gray ice can be falsely classified as a dark lead in ICESat-2 data. New/gray ice being treated as open ocean will lower the LIF calculated from ICESat-2, and is therefore a critical consideration for the results and comparisons presented in the manuscript. I was disappointed not to see any discussion on this. How consistently is new/gray ice classified as ice rather than open ocean in ATL07? Have the authors verified this with multiple scenes? Do they expect any significant implications of incorrectly classifying new/gray ice as a lead in LIF calculation? The inclusion of an LIF (specular) product does not address the intricacies of the problem.

While we agree that IS-2 validation is important, it is not the stated goal of our work and it remains a necessary objective for the IS-2 community. We add citations that point to these validative works and an additional discussion of the challenge of grey/non-grey ice in both Parts.

Specific comments:

L40: I don't think it's technically true that ICESat-2 can resolve leads at the submeter scale. The ICESat-2 footprint is stated to be 10 m in the manuscript, and the resolution of the ATL07 data is even more coarse, which is the value that's applicable for this manuscript.

We removed this sentence as it was confusing and address IS-2 resolution earlier.

L60: Considering the large number of scenes analyzed, what fraction are visually validated? Based on that, can this really be considered a "validation"?

We discuss the visual validation performed in the preceding article.

L65: It isn't clear what's meant by "equal to the maximum of the Bootstrap and NASATeam algorithms"

We added a longer discussion of PM products in which we talk about the NSIDC algorithm.

Section 2, in general, was difficult to follow. For example, there's a lot of jumping around between tenses when explaining the method. L61-73 especially would confuse someone who's not very familiar with each product. I'd suggest re-wording how the products are introduced.

We have rewritten this section.

L90: I'm not sure "worst-performing" is the best phrase here. At present there's not enough in the text to convince me it's the worst-performing, based purely on having the greatest differences with OIB.

We agree and change this text.

L32: I'd suggest also considering data quality flags

We do consider these in the product generation but omitted this here, we add now.

Figure 3: I assume the 3 ICESat-2 lines show just the strong beams? State this in the figure caption, considering the analysis is for all 6 beams.

All 6 are shown but beam pairs are near relative to the image size. We add description to the text.

L160: Do you mean April 7, 2022 rather than May?

This is an image on May 7, 2022- see table 2. We add (Table 2) in the text.

I'm struggling with the description of the product as "global", when in the Arctic it

only covers 25-65% of the sea ice zone

We expanded the discussion of "global" as an adjective and do not use it when it may give the impression of a more extensive product.

State what release of the ICESat-2 ATL07 data you're using. If you're not using release 6 (the latest version of the data), please explain the reason for that.

Indeed, we use v6 and will include a mention in the revised manuscript.

Technical comments

L48: Change ICESat-2 to IS2 L176: Change if to is L58: Define DMS L61: Acronym has not been defined yet L104: Random italic font for "overestimate" L106: Change unit to integer L123: IS2 has already been defined. Check throughout. L218: Change cell to cells

All to be addressed in the revision.

Refs

Bröhan, D.; Kaleschke, L. A Nine-Year Climatology of Arctic Sea Ice Lead Orientation and Frequency from AMSR-E. Remote Sens. 2014, 6, 1451-1475. https://doi.org/10.3390/rs602145